# Cooperation between distinct viral variants promotes growth of H3N2 influenza in cell culture

Katherine S Xue[1,2], Kathryn A Hooper[2,3], Anja R Ollodart[2,3], Adam S Dingens[2,3], Jesse D Bloom[1,2]*

[1]Department of Genome Sciences, University of Washington, Seattle, United States; [2]Division of Basic Sciences, Fred Hutchinson Cancer Research Center, Seattle, United States; [3]Molecular and Cellular Biology Program, University of Washington, Seattle, United States

**Abstract** RNA viruses rapidly diversify into quasispecies of related genotypes. This genetic diversity has long been known to facilitate adaptation, but recent studies have suggested that cooperation between variants might also increase population fitness. Here, we demonstrate strong cooperation between two H3N2 influenza variants that differ by a single mutation at residue 151 in neuraminidase, which normally mediates viral exit from host cells. Residue 151 is often annotated as an ambiguous amino acid in sequenced isolates, indicating mixed viral populations. We show that mixed populations grow better than either variant alone in cell culture. Pure populations of either variant generate the other through mutation and then stably maintain a mix of the two genotypes. We suggest that cooperation arises because mixed populations combine one variant's proficiency at cell entry with the other's proficiency at cell exit. Our work demonstrates a specific cooperative interaction between defined variants in a viral quasispecies.

*For correspondence: jbloom@fhcrc.org

**Competing interests:** The authors declare that no competing interests exist.

## Introduction

The evolution of RNA viruses is characterized by high mutation rates and large population sizes, which together create genetically diverse populations known as quasispecies (*Eigen, 1971*; *Holland et al., 1982*; *Lauring and Andino, 2010*; *Andino and Domingo, 2015*). High levels of standing genetic diversity can provide a substrate for selection and rapid adaptation, an advantage for viruses that experience strong and varied selective pressures to escape immune recognition, develop drug resistance, and adapt to new hosts (*Najera et al., 1995*; *Pfeiffer and Kirkegaard, 2005*; *Dutta et al., 2008*).

Recently, several studies have suggested that cooperative interactions between variants in a quasispecies can also increase population-level fitness (*Vignuzzi et al., 2006*; *Ciota et al., 2012*; *Shirogane et al., 2012*; *Ke et al., 2013*; *Bordería et al., 2015*). *Vignuzzi et al., 2006* found that genetically diverse poliovirus populations were required for wild-type neurotropism and pathogenesis, leading the authors to suggest that unknown cooperative interactions among minor variants promoted the overall fitness of the population. More recent studies have suggested cooperation between distinct variants of measles (*Shirogane et al., 2012*), hepatitis B virus (*Cao et al., 2014*), and Coxsackie virus (*Bordería et al., 2015*). However, specific examples of robust cooperative interactions between defined variants in viral quasispecies remain rare (*Holmes, 2010*).

Here, we demonstrate that cooperation between two distinct variants of human H3N2 influenza promotes viral growth in cell culture. The two variants differ by a single amino-acid mutation in the neuraminidase (NA) protein, which normally mediates viral exit from the host cell. Both variants are

**eLife digest** Viruses like influenza mutate fast. When you get the flu, the virus hijacks your cells, and your body becomes home to millions of viruses, many of which are genetically different from each other. Previous research had suggested that variants of rapidly evolving viruses sometimes cooperate with one another to survive, but few studies have pinpointed specific cooperative interactions.

In the past decade, influenza surveillance groups noticed that one particular mutant virus appears again and again when influenza viruses are grown in the laboratory. Xue et al. thought that this mutant virus shouldn't be able to grow on its own because the mutation disrupted the protein that influenza viruses use to detach from host cells. So, they asked if the mutant was cooperating with the non-mutated virus to survive.

Xue et al. revealed that the two influenza viruses, which differ by just one mutation, cooperate with each other when grown in cells in the laboratory. The mutant virus often appeared following a random mutation in a population of non-mutated virus, and vice versa. Instead of competing with each other until one virus went extinct, the two viruses actually grew better together than they did apart. Xue et al. suggest that this is because one of the viruses is good at entering new cells, while the other is better at exiting cells to spread the infection. A mixed population combines these two strengths.

Following on from this work, it remains unclear whether influenza viruses cooperate in other settings – for example, during infections in people. Further studies are also needed to determine exactly how the two viruses help each other at the molecular level.

reported numerous times in human H3N2 NA sequences deposited in the GISAID EpiFlu database, and both have been observed in mixed populations when clinical specimens are passaged in cell culture. We show that the two variants grow better together than apart, and that serial passage repeatably selects for mixed populations. We suggest that the cooperation arises because one variant is proficient at cell entry while the other is proficient at cell exit. Overall, our work represents a clear example of selection to generate and maintain two cooperating genotypes within a viral quasispecies.

# Results

## Mutations at site 151 in H3N2 neuraminidase tend to occur in mixed populations

Over the last decade, several groups have reported that mutations arise rapidly and repeatedly at residue NA 151 when human H3N2 influenza is passaged in cell culture (*Table 1*) (*McKimm-Breschkin et al., 2003*; *Lin et al., 2010*; *Tamura et al., 2013*; *Lee et al., 2013*; *Chambers et al., 2014*; *Mishin et al., 2014*; *Mohr et al., 2015*). Residue 151 is in the NA active site and is highly conserved; until recently, it had an amino-acid identity of D in virtually all N2 NAs. Ordinarily, NA mediates viral exit from the host cell by cleaving sialic-acid receptors to release newly produced virions. The D151G mutation ablates the catalytic activity of NA and instead causes it to bind the receptors that it typically cleaves (*Zhu et al., 2012*). Mutations at this site seem to be more common in viruses that have been passaged in cell culture compared to the original clinical isolates (*Deyde et al., 2009*; *Lin et al., 2010*; *Okomo-Adhiambo et al., 2010*; *Tamura et al., 2013*; *Lee et al., 2013*; *Chambers et al., 2014*; *Mishin et al., 2014*). As a result, mutations at site 151 have been categorized as lab adaptations (*Okomo-Adhiambo et al., 2010*; *Tamura et al., 2013*; *Lee et al., 2013*; *Mishin et al., 2014*).

We examined whether mutations at site 151 exhibited patterns consistent with simple lab adaptation by determining the frequencies of amino acids at this position in human H3N2 NA sequences in the GISAID EpiFlu database for each year from 2000 to 2014 (*Figure 1*). Most isolates in this database are first passaged in eggs or cell culture, and then the consensus sequence of the viral population is determined by Sanger sequencing. Beginning in 2007, the frequency of mutations at NA site

**Table 1.** Prior reports of variation at neuraminidase site 151 when H3N2 clinical specimens are passaged in cell culture.

| Reference | Summary |
| --- | --- |
| (*McKimm-Breschkin et al., 2003*) | Sanger sequencing of 38 oseltamivir- and zanamivir-resistant MDCK-passaged clinical isolates found that 7 had G, N, E, or V at site 151. |
| (*Lin et al., 2010*) | Sanger sequencing of 18 isolates after passage in MDCK or MDCK-SIAT1 cells found 4 isolates as D+G, 3 as D+N, and 2 as D+A. Pyrosequencing detected low frequencies of G151 and N151 in some clinical samples. |
| (*Tamura et al., 2013*) | Pyrosequencing of 150 isolates after 1-4 passages in MDCK cells found that 85% developed mixed populations at site 151; 29% did so after a single passage. Mixed populations consisted of D+N, D+G, D+G+N, and D+G+A genotypes. T148I/K/P mutations were also observed in 23% of isolates. |
| (*Lee et al., 2013*) | 77 clinical specimens were Sanger-sequenced before and after a single passage in MDCK cells. 18 acquired a mutation at NA site 151: 10 were D+N, 7 were D+G and one fixed D151N at the limit of detection. No mutations were detected in the unpassaged specimens. |
| (*Chambers et al., 2014*) | 9 A/Victoria/361/11-like clinical specimens were passaged twice in MDCK cells and Sanger-sequenced before and after expansion. 4 isolates developed NA-dependent cell binding; 3 had D151G, the other D151N. |
| (*Mishin et al., 2014*) | Pyrosequencing of 150 MDCK-grown isolates found that 42 were D+G, 34 were D+N, and 57 were D+G+N. Pyrosequencing of 50 matched clinical specimens detected no variation at site 151. |
| (*Mohr et al., 2015*) | 16 pairs of isolates cultured in parallel in MDCK cells and in eggs were sequenced using Ion Torrent. 5 MDCK isolates were D+N, 4 were D+G, and 2 were D+N+G. No egg-passaged isolates had mutations at site 151. T148I/K mutations were observed in 7 MDCK isolates. |

151 rose dramatically, with mutant genotypes representing about a quarter of the sequences. G151 and N151 are each reported in about 1% of sequences, but ambiguous nucleotide calls at the codon make up the majority of non-wild-type sequences. Because these sequences usually represent consensus calls from Sanger sequencing, the ambiguous nucleotides likely indicate the presence of mixed D151+G151 and D151+N151 populations. The relative abundance of mixed populations in strains deposited in the GISAID EpiFlu database is consistent with the fact that, in tissue culture, mutations at site 151 arise frequently but fix rarely (*Table 1*) (*Lin et al., 2010*; *Okomo-Adhiambo et al., 2010*; *Tamura et al., 2013*; *Lee et al., 2013*; *Mishin et al., 2014*; *Mohr et al., 2015*).

When viruses are passaged, they experience culture-specific selective pressures. To understand how variation at site 151 might depend on the passaging procedures, we classified sequences according to their passage histories as annotated in the GISAID EpiFlu database. Prior to 2007, site 151 is almost always reported to be D, regardless of passage history (*Figure 1B*). From 2007 onward, variants at site 151 are reported almost exclusively in isolates that have been passaged in cell culture (*Figure 1C*). Whereas nearly a third of isolates that have been passaged in cell culture are reported to have an amino acid other than D at site 151, only two of nearly 1800 unpassaged isolates and five of nearly 400 egg-passaged isolates are reported to have non-D amino acids at site 151. These observations accord with reports in the literature that mutations at site 151 are observed primarily in cell-culture-passaged isolates (*Tamura et al., 2013*; *Lee et al., 2013*; *Mohr et al., 2015*). However, it is important to remember that the methods used to determine most sequences in the GISAID EpiFlu database lack sensitivity to detect minority variants in a viral population. For instance, the experimental results that we describe below suggest that it is exceedingly unlikely that any of the more than 100 reported G151 sequences actually reflect the complete fixation of this mutation.

Overall, the results in *Figure 1* indicate that the G151 mutation tends to occur in mixed populations. We therefore sought to experimentally characterize the growth of pure D151 and G151 viral variants to determine whether mixed populations represent incomplete fixation of a lab-adaptation mutation or whether they are the product of active selection.

## Mixed populations of D151 and G151 viruses grow better than pure populations of either variant

We compared the growth of the D151 and G151 variants both alone and in mixed populations by generating viruses of defined genotypes using reverse genetics. The A/Hanoi/Q118/2007 strain, henceforth referred to as Hanoi/2007, is a human H3N2 strain with a G151 genotype. Its NA protein

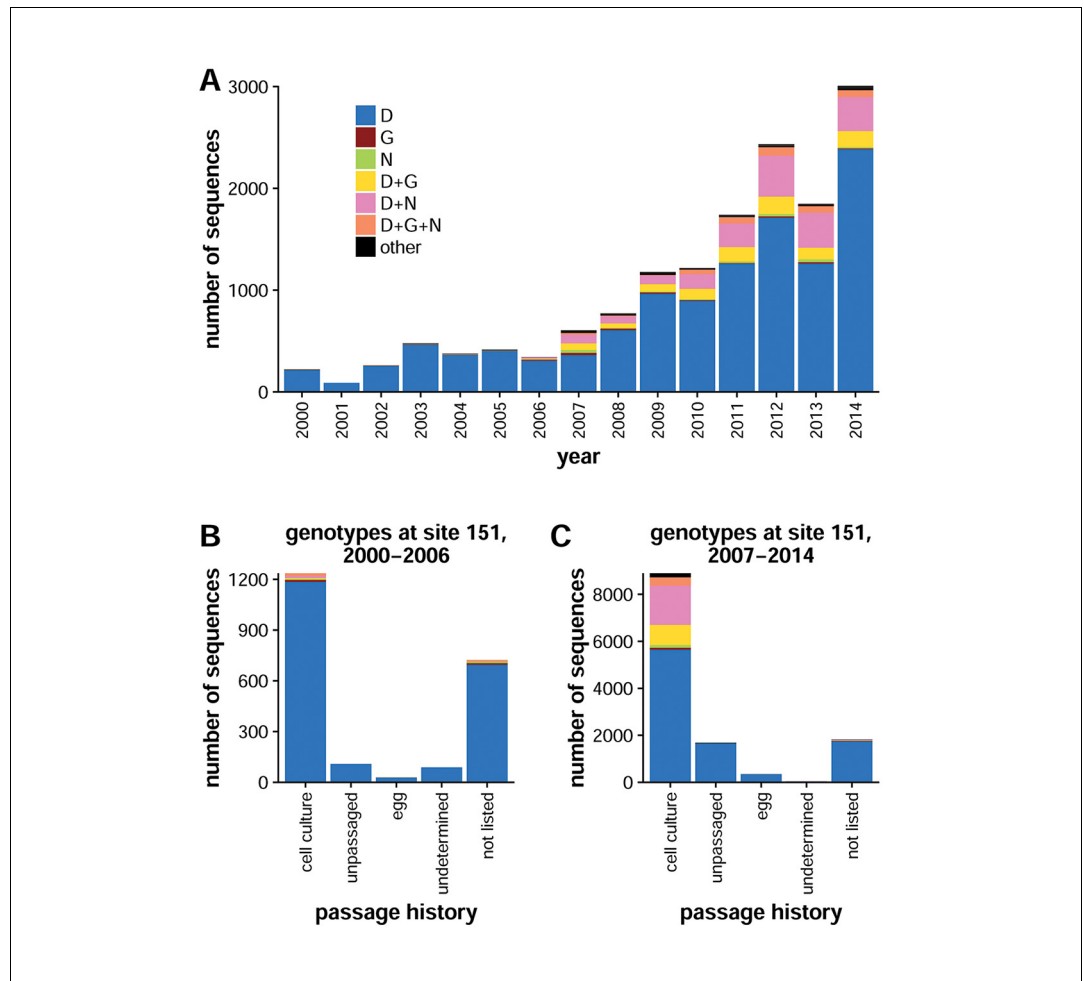

**Figure 1.** Ambiguous identities are common at NA site 151 after 2007. (**A**) Shown are the number of human H3N2 influenza NA sequences in the GISAID EpiFlu database with the given identity at site 151 for each year from 2000 to 2014. Since 2007, ambiguous amino-acid identities have been present at residue 151 in about 20% of sequences. Sequences from (**B**) 2000 to 2006 and (**C**) 2007 to 2014 were classified into groups based on their passage history. Ambiguous amino-acid identities were present almost exclusively in isolates that had been passaged in cell culture. Sequences were classified as 'undetermined' if the passage history was difficult to interpret and as 'not listed' if the passage history was absent altogether. Mixed genotypes were inferred on the basis of IUPAC nucleotide ambiguity codes; for instance, the triplet GRT could refer to GAT or GGT, corresponding to amino acids D and G, respectively. Genotypes are indicated if they exceeded a frequency of 0.5% among all analyzed sequences; otherwise, they are categorized as 'other.' The computer code used for analysis is available in *Figure 1—source data 1*.

The following source data is available for figure 1:

**Source data 1.** This 7-zip archive contains the source code used for *Figure 1* (the analysis of mutation frequencies at site 151 in naturally occurring sequences).

sequence is identical to that of several other sequenced isolates, except that the other strains have D at site 151. We created reverse-genetics plasmids encoding the protein sequences of both the D151 and G151 variants of the Hanoi/2007 NA, as well as the HA from this strain. The internal genes were derived from the lab-adapted A/WSN/33 influenza strain with GFP packaged in the PB1 segment (**Bloom et al., 2010**). The two viral variants were therefore isogenic except for the variation at site 151.

We generated virus using reverse genetics by co-transfecting cells with plasmids encoding the D151 variant, the G151 variant, or an equal mix of the two, together with isogenic plasmids for the

other viral genes, and we quantified the resulting titers (*Figure 2A*). Surprisingly, given its wide-spread designation as a lab adaptation, the G151 variant grows extremely poorly. However, a mixed population of D151 and G151 variants grows to substantially higher titers than the corresponding pure population of D151 viruses. The growth advantage of the mixed population suggests that cooperation between the two variants improves viral growth.

We next sought to determine whether there was also a cooperative effect when the D151 and G151 variants were mixed in direct infections, since generation of influenza virus by reverse genetics is a complex process that involves co-transfecting cells with plasmids encoding each of the eight viral genes. We generated pure populations of D151 and G151 viruses by reverse genetics, growing both populations in the presence of 50 nM oseltamivir, a small molecule that competes with sialic acid for binding to the NA active site. The addition of oseltamivir increases the titers of the G151 variant (*Figure 2—figure supplement 2*) and presumably prevents selection for *de novo* NA mutations by suppressing both the cleavage and binding activity of this protein. We then infected cells with pure D151 viruses, pure G151 viruses, or an equal mix of both variants at a total multiplicity of infection (MOI) of 0.2. One hour post-infection, we washed the cells to remove residual oseltamivir and then monitored viral replication. These experiments were performed in full biological triplicate, beginning with triplicate independent creations of each pure population by reverse genetics.

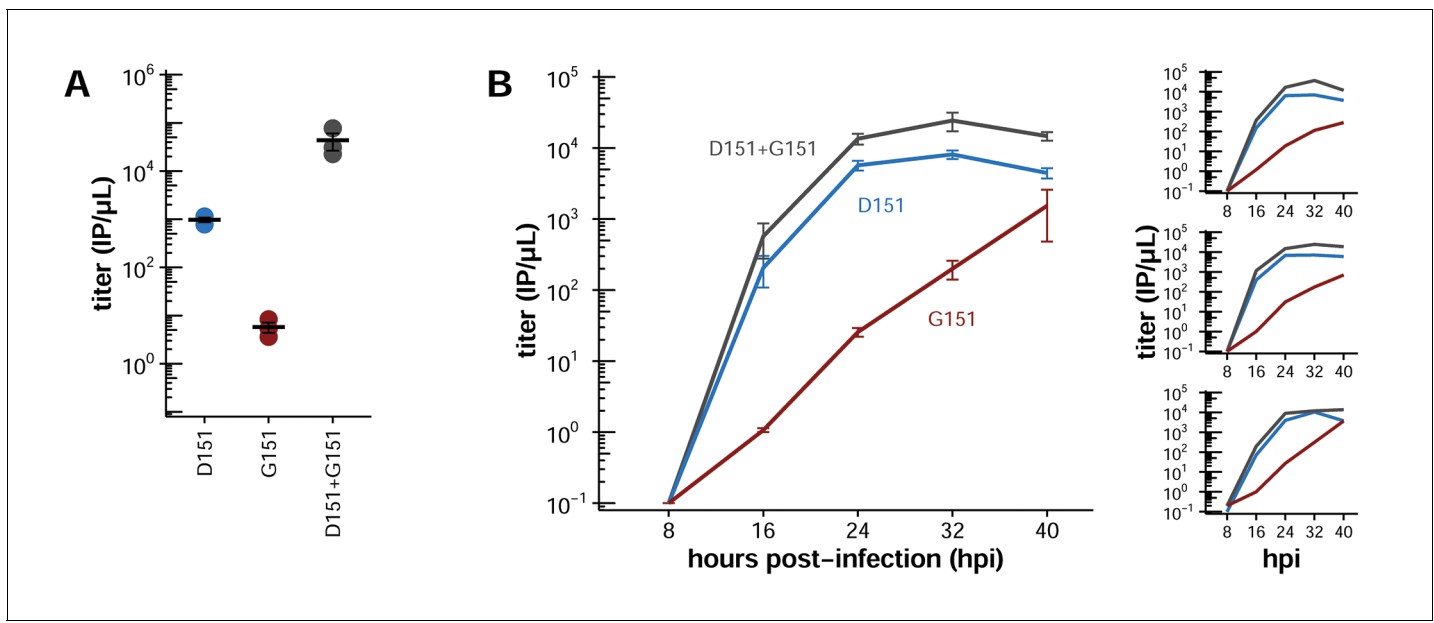

**Figure 2.** Mixed populations grow to higher titers than either pure population alone. (**A**) Pure and mixed populations were generated by reverse genetics. Cells were transfected with a Hanoi/2007 NA plasmid encoding D151, G151, or an equal mix of the two, along with isogenic plasmids for the other genes. The total amount of NA plasmid was the same in all cases; that is, the pure populations were transfected with 250 ng of the indicated variant, and the mixed populations were transfected with 125 ng of each variant. The HA was also derived from Hanoi/2007, and the other genes were derived from the lab-adapted A/WSN/33 strain with GFP packaged in the PB1 segment. The titer was determined after 72 hr using the GFP reporter. Black lines indicate the mean and standard error of the titers for three biological replicates, with titers for each replicate plotted as points. *Figure 2—figure supplement 1* shows a comparable effect when the virus does not package GFP. *Figure 2—figure supplement 2* shows that growth of the G151 variant is improved by adding oseltamivir. (**B**) Cells were infected at an MOI of 0.2 with pure D151 virus, pure G151 virus, or an equal mix of the two. The total MOI of infecting virus was the same in all cases. The main plots show titers averaged across three biological replicates, with each replicate plotted individually in the small insets.

The following figure supplements are available for figure 2:

**Figure supplement 1.** A mixed population outgrows either pure population when viruses are generated by reverse genetics with an unmodified PB1 gene.

**Figure supplement 2.** Growth of the G151 variant is improved by adding oseltamivir during the generation of viral populations by reverse genetics.

Once again, the mixed populations consistently grew more rapidly and reached higher maximal titers than either pure population (*Figure 2B*). The trends in the direct co-infections were similar to those observed when generating the viruses by reverse genetics. The pure G151 populations grew very poorly, again showing that this variant has very low fitness on its own. The pure D151 populations grew reasonably well on their own, but the mixed populations grew even better. These results show that cooperation between the D151 and G151 variants improves growth of the overall population.

Interestingly, viral titers increased sharply late in the passage in some G151 populations. One possibility is that *de novo* mutations to the D151 variant create a mixed population with higher fitness. To explore the possibility of *de novo* emergence of cooperation, we serially passaged pure and mixed populations as described below.

## Serial passage selects for mixed populations of D151 and G151 viruses

If the D151 and G151 variants cooperate, then we expect mixed populations to emerge by *de novo* mutation and to be stably maintained when they already exist. To test this prediction, we serially passaged pure and mixed viral populations and performed targeted deep sequencing of the NA gene at the end of each passage to assess changes in allele frequency at site 151. We again used reverse genetics to generate triplicate pure populations of D151 and G151 viral variants in the presence of 50nM oseltamivir, then infected cells with D151 viruses, G151 viruses, or an equal mix of the two at a total MOI of 0.2, washing the cells one hour post-infection to remove residual oseltamivir. We verified that the D151 and G151 populations used to inoculate the first passage were pure within our limit of detection of approximately 1%, which we determined by deep-sequencing pure plasmid. We performed a total of five serial passages for each replicate, in each case seeding the new passage with the supernatant from the previous one at a total MOI of 0.2.

The mixed D151+G151 populations maintained an approximately equal mix of the two variants through all five passages (*Figure 3*). In the pure populations, the opposite variant arose by *de novo* mutation, then rose in frequency as the population converged towards a roughly equal mix of the two variants. The D151 variant emerged rapidly during passage of the G151 populations, exceeding a frequency of 20% by the end of the second passage in all three replicates. The G151 variant was slower to arise in the D151 populations but had reached a substantial frequency by the end of passage 4 in all three replicates. The changes in allele frequency during serial passage demonstrate that selection acts to balance the proportion of these two genotypes in the population.

In one of the D151 populations, N151 also emerged spontaneously, and by the end of passage 5, the population consisted of a mix of D151, N151, and G151. Like G151, N151 commonly occurs in mixed populations with D151 in sequences in the GISAID EpiFlu database (*Figure 1*) and is mentioned in reports of mutations at site 151 in cell culture (*Table 1*) (*McKimm-Breschkin et al., 2003*; *Lin et al., 2012*; *Okomo-Adhiambo et al., 2010*; *Tamura et al., 2013*; *Lee et al., 2013*; *Chambers et al., 2014*; *Mishin et al., 2014*; *Mohr et al., 2015*). We verified that N151 cooperates with D151 by creating the N151 variant of the Hanoi/2007 NA and generating pure and mixed populations by reverse genetics (*Figure 3—figure supplement 1*). N151 viruses behave similarly to G151 viruses: they grow very poorly on their own, but cooperate with D151 to outgrow either pure population.

These results show that serial passage selects for mixed populations of D151 and G151 variants, even when the starting population is isogenic. Furthermore, mixed populations are stably maintained; the G151 variant does not sweep to fixation, as would be expected for a simple lab adaptation. Cooperation between the D151 and G151 variants evidently selects for the generation and maintenance of a genetically diverse quasispecies.

## The dynamics of cooperation depend on the multiplicity of infection

Since each influenza virion typically packages only a single copy of the NA gene, co-infection of a cell by multiple viruses is likely to increase opportunities for interactions among viral variants. *Figure 2* shows that at an MOI of 0.2, the mixed populations of D151 and G151 variants have an advantage over pure populations. At a lower MOI, co-infection is less likely. We therefore sought to test whether cooperation also promotes growth at higher and lower MOIs. We infected cells with pure

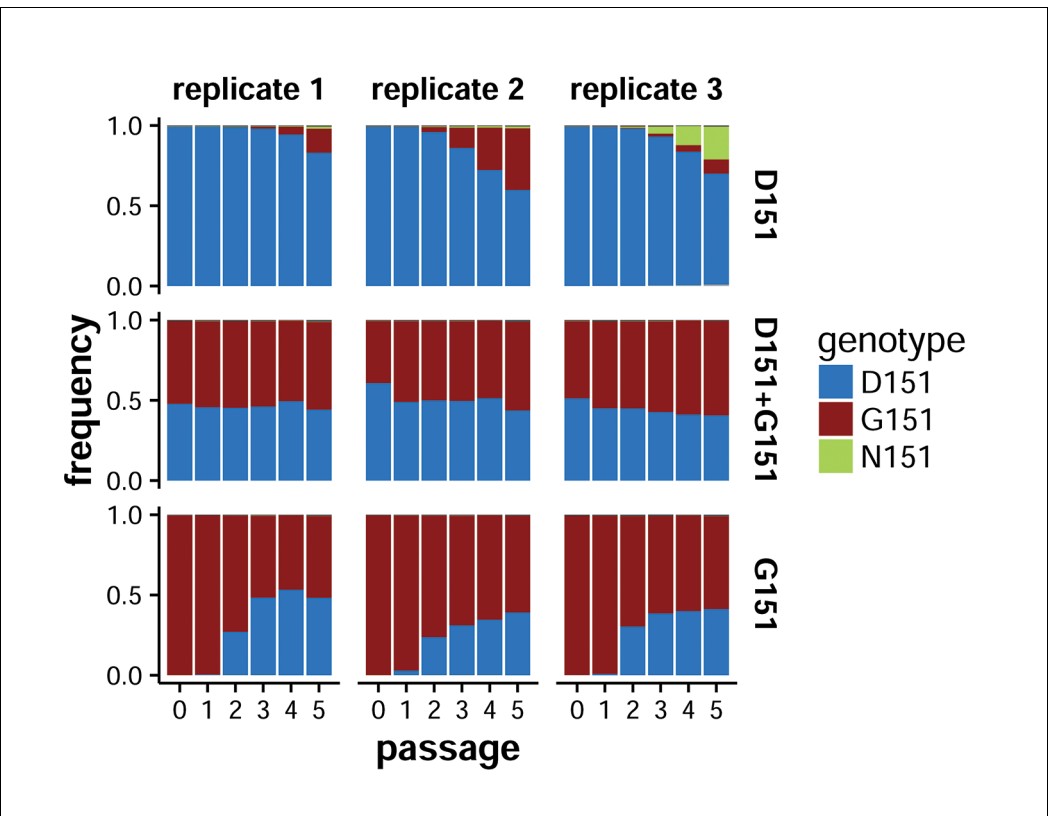

**Figure 3.** Serial passage selects for a stable mix of the two variants. Shown are the allele frequencies at NA site 151 over five tissue-culture passages of initially pure D151 viruses, pure G151 viruses, or an equal mix of the two. Each passage was seeded at a total MOI of 0.2. Passage 0 refers to the ratio of variants in the viral inoculum for passage 1. Allele frequencies were determined by targeted Illumina deep-sequencing of the NA gene. Based on sequencing of pure plasmid, the error rate was less than 1%. The raw data and computer code are available in *Figure 3—source data 1*.

The following source data and figure supplement are available for figure 3:

**Source data 1.** This 7-zip archive contains the data and source code used for *Figure 3* (the analysis of mutation frequencies at site 151 after serial passage in the lab).
**Figure supplement 1.** The N151 variant also cooperates with D151.

and mixed viral populations in biological triplicate at an MOI of 0.02 and an MOI of 0.5, and then monitored viral titers over the next 40 hr as in *Figure 2*.

At an MOI of 0.02, the mixed populations grew similarly to or slightly worse than the D151 populations for the first 24 hr post-infection (*Figure 4A*). Later in the infection, however, the mixed populations grew to substantially higher titers than the D151 populations. In contrast, at MOIs of 0.2 (*Figure 2B*) and 0.5 (*Figure 4B*), the mixed populations grew better than the pure populations throughout the entire infection.

We note that the effective MOI of an infection increases as the infection progresses as newly produced viruses accumulate in the supernatant (*Wilke et al., 2004*). For the infections inoculated at an MOI of 0.02, the sharp increase in titers for the mixed population late in the infection are likely a result of this higher effective MOI. We therefore conclude that the dynamics of cooperation depend on the multiplicity of infection, with the cooperative effect decreased at lower MOIs.

## Changes in HA potentiated cooperation between the NA variants

Mutations at NA site 151 become common in the EpiFlu database only starting in 2007 (*Figure 1A*), suggesting that other mutations to the influenza genome around that date might have affected the

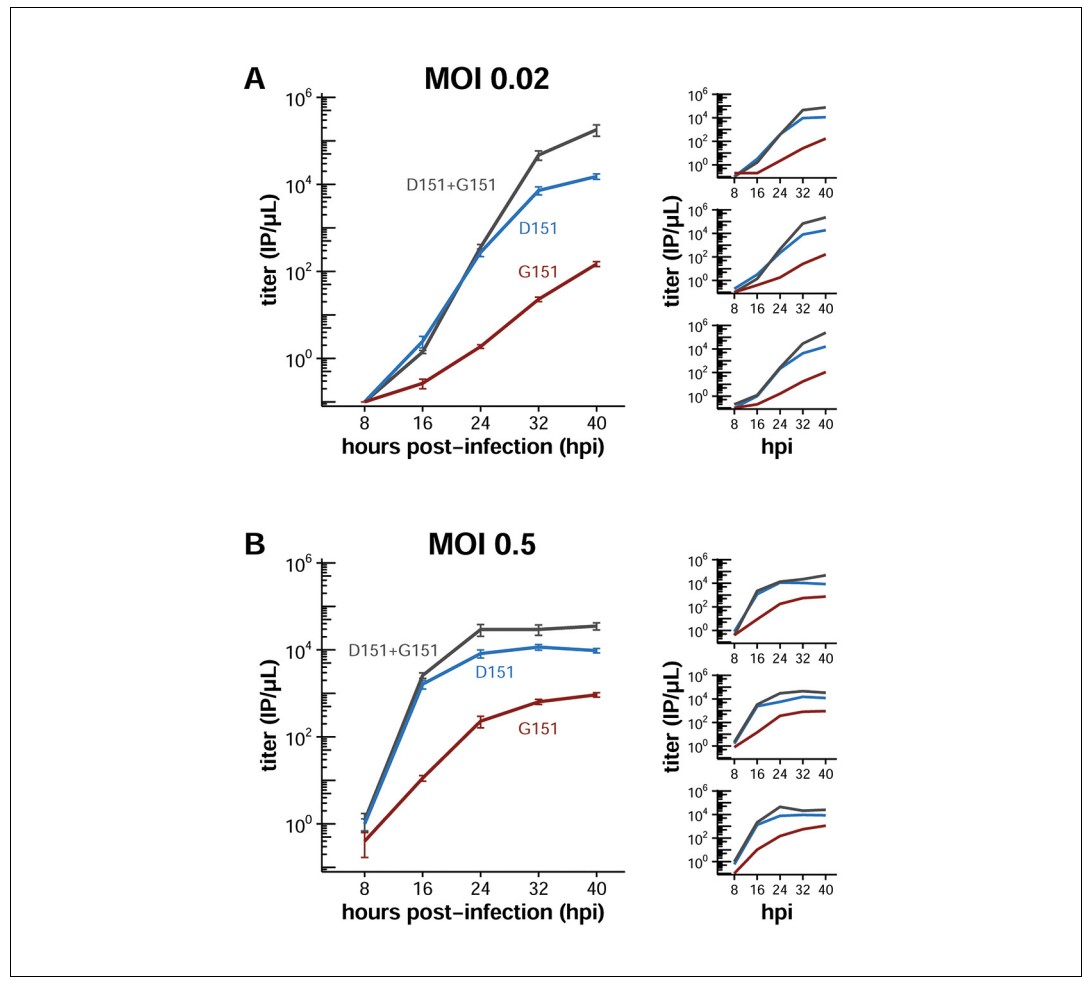

**Figure 4.** Cooperative dynamics depend on multiplicity of infection. Cells were infected at an MOI of (**A**) 0.02 or (**B**) 0.5 with pure D151 virus, pure G151 virus, or an equal mix of the two. The total MOI of infecting virus was the same across the mixed and pure populations for infection at each MOI. The main plots show titers averaged across three biological replicates, with each replicate plotted individually in the small insets. The experiments here parallel those in **Figure 2B**. Black lines indicate the mean and standard error of the titers for three biological replicates, with titers for each replicate plotted as points.

potential for cooperation among NA variants at site 151. A candidate gene for these potentiating mutations is HA. Good viral growth requires a balance between the receptor binding of HA and the receptor cleaving of NA (*Wagner et al., 2002*; *Gulati et al., 2005*; *Neverov et al., 2015*). For reasons that remain unclear, the HAs of recent human H3N2 influenza have lost their affinity for many types of sialic acid (*Lin et al., 2012*; *Gulati et al., 2013*). We therefore hypothesized that recent mutations in HA might have potentiated cooperation by making it advantageous for viral populations to acquire the NA-mediated receptor-binding of the G151 variant (*Zhu et al., 2012*) to compensate for reduced HA binding.

To test this hypothesis, we examined the effects of the D151 and G151 NA variants in viruses that had the HA of an earlier H3N2 strain, A/Wisconsin/67/2005, henceforth referred to as Wisconsin/2005. We cloned the HA gene from the Wisconsin/2005 strain into a reverse-genetics plasmid and generated pure and mixed populations of D151 and G151 variants in the genetic background of either the Hanoi/2007 HA or the Wisconsin/2005 HA. Cooperation between the D151 and G151 variants was eliminated in the Wisconsin/2005 HA background (*Figure 5*). Therefore, some of the changes to HA that distinguish the Wisconsin/2005 and Hanoi/2007 homologs are important for potentiating cooperation between the NA variants.

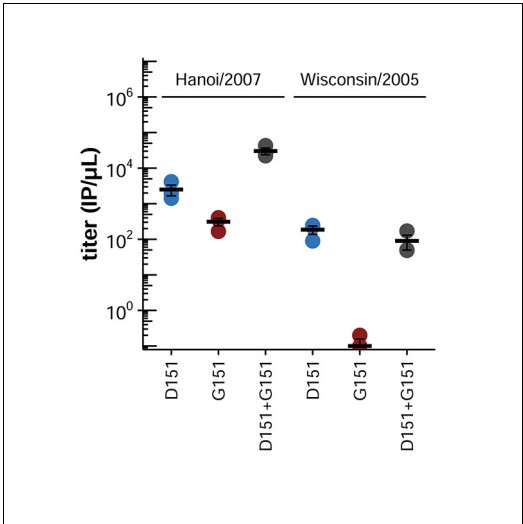

**Figure 5.** Changes in HA between 2005 and 2007 potentiated cooperation. Cooperation occurs between the D151 and G151 NA variants in viruses with HA from the Hanoi/2007 strain, but not in viruses with HA from the Wisconsin/2005 strain. Shown are the titers after reverse genetics with the indicated HA and NA. The experiments here parallel those in *Figure 2A*. Black lines indicate the mean and standard error of the titers for three biological replicates, with titers for each replicate plotted as points.

If decreased HA receptor-binding potentiates cooperation between the receptor-cleaving D151 and receptor-binding G151 NA variants, then viral growth should depend entirely on this cooperation if NA is the only protein able to bind the receptor. To test this hypothesis, we used an HA that has been heavily engineered to eliminate its receptor-binding activity (*Hooper and Bloom, 2013*). We used reverse genetics to generate pure and mixed populations of D151 and G151 NA variants paired with this binding-deficient HA, and we measured viral titers (*Figure 6*). In the absence of HA receptor binding, neither the D151 nor the G151 variant alone reached appreciable titers. However, the mixed population was still able to grow with the binding-deficient HA. These results show that cooperation becomes obligate in the absence of HA receptor binding, presumably because NA must serve as the sole source of both binding and cleaving.

## Discussion

We have shown that cooperation between two distinct variants of human H3N2 influenza promotes viral growth in cell culture. These variants differ by a single amino-acid mutation in NA, and each variant is present in many human H3N2 isolates that have been analyzed by Sanger sequencing after passage in the lab. Prior work has assumed that the less common G151 variant is a lab-adaptation mutant that emerges as the more common D151 variant is passaged in cell culture. Our work shows, however, that evolution in cell culture selects for a balanced mix of both variants. The G151 variant can barely replicate alone, but it cooperates with the D151 variant to increase population fitness. After multiple serial passages, both pure and mixed populations converge to an equilibrium in which both variants are present at approximately equal frequencies. Our work therefore represents a clear example of cooperation between distinct variants in a viral quasispecies.

We propose that cooperation arises because one variant is proficient at cell entry, while the other is proficient at cell exit. Viruses with wild-type D151 NAs always exit cells efficiently, since their NAs cleave sialic-acid receptors to facilitate viral release. However, the HAs of recent human H3N2 strains have reduced affinity for many sialic-acid receptors (*Lin et al., 2012*; *Gulati et al., 2013*), reducing the efficiency with which those viruses can attach to many cells via HA. G151 viruses are proficient at cell entry, since their NA binds strongly to sialic acid, but they cannot detach effectively from host cells due to a lack of catalytic activity (*Zhu et al., 2012*). But in combination, D151 and G151 enable both efficient cell exit and entry. Indeed, our experiments with a binding-deficient HA indicate that in a mixed D151 and G151 population, NA can act as the exclusive source of both receptor binding and receptor cleaving (*Figure 6*). Our results evoke prior work showing that fitness in a Coxsackie virus population is enhanced by the combination of multiple receptor-binding variants (*Bordería et al., 2015*).

How do the D151 and G151 variants collaborate to enable both viral entry and exit at the level of individual virions? Co-infection of the same cell with both D151 and G151 variants would produce progeny that have both NA variants on their surface, even though each new virion would package only a single copy of the NA gene. We suspect that much of the observed cooperation may result from co-infections that produce such mixed-NA virions, which would then carry proteins that make them proficient at both cell entry and cell exit. We found that MOI affects cooperative dynamics, supporting this interpretation (*Figure 4*). However, other mechanisms could also contribute. In a mixed population,

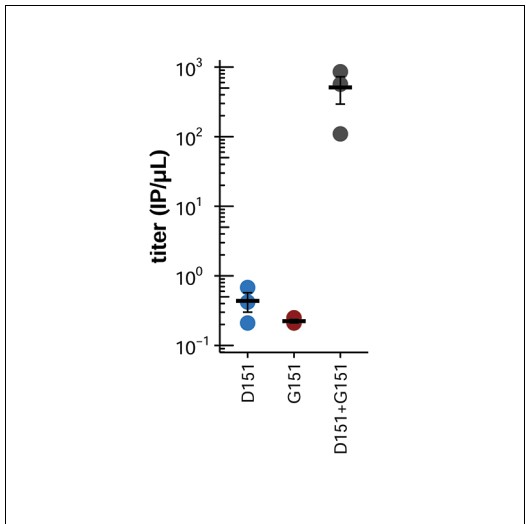

**Figure 6.** Cooperation is obligate when HA lacks receptor-binding activity. Shown are the titers after reverse genetics with the indicated Hanoi/2007 NAs in combination with an engineered binding-deficient H3 HA with multiple mutations to the receptor-binding pocket (**Hooper and Bloom, 2013**). The experiments here parallel those in **Figure 2A**. Black lines indicate the mean and standard error of the titers for three biological replicates, with titers for each replicate plotted as points.

D151 viruses may cleave G151 viruses from the cell surface without both protein variants being present on the same infectious particle, since sialidase activity can promote viral growth in *trans* (**Liu and Air, 1993**). More detailed molecular characterization of virions in mixed populations will be necessary to establish the exact mechanism of cooperation.

It remains unclear whether the D151 and G151 variants cooperate in clinical infections. When we analyzed the passage histories of sequenced isolates, we found that mixed populations were reported almost exclusively in isolates that had been passaged in cell culture (**Figure 1C**). Several groups have reported that mutations emerge at NA site 151 when clinical isolates are expanded in cell culture – but with one exception (**Lin et al., 2010**), Sanger sequencing or pyrosequencing of matched clinical and passaged isolates has so far failed to detect variation in site 151 in unpassaged isolates (**Lee et al., 2013**; **Chambers et al., 2014**; **Mishin et al., 2014**) (see also **Table 1**). However, the sequencing methods used by these studies are relatively insensitive to low-frequency variation. Given how quickly and frequently D151 mutations sometimes arise—one group found that nearly a quarter of isolates showed variation at appreciable frequencies after a single passage in MDCK cells (**Lee et al., 2013**)—pre-existing variation at site 151 in the original clinical isolates could contribute to the observed evolution. More sensitive deep sequencing of unpassaged clinical isolates will be necessary to resolve these questions.

Our work demonstrates that cooperation between distinct viral variants can enhance the population's overall fitness. This cooperation is not a rare event; the cooperating variants that we describe emerge rapidly and repeatedly, both in our own experiments and apparently in hundreds of clinical isolates passaged by numerous labs. Our work emphasizes that genetic diversity in viral populations can be more than a transient state that facilitates adaptation: it can itself be a beneficial trait that is generated and maintained by selection. As the deep sequencing of viruses becomes increasingly common in microbiology and epidemiology, it will be important to better understand the broader role that cooperation plays in the evolution and maintenance of population-level diversity.

## Materials and methods

### Analysis of GISAID EpiFlu sequences

We downloaded the set of 15,079 sequences in the Global Initiative on Sharing All Influenza Data (GISAID) EpiFlu database (**Bogner et al., 2006**) corresponding to all full-length NA coding regions from human H3N2 influenza A isolates collected from January 1, 2000 to December 31, 2014. We pairwise aligned each sequence to the A/Hanoi/Q118/2007 (H3N2) coding sequence (Genbank accession CY104446) using the program needle from EMBOSS version 6.6.0 (**Rice et al., 2000**), which implements a Needleman-Wunsch alignment. We identified the genotype of each sequence at site 151 and parsed the sequence metadata to determine the year in which it was collected and the sequence's passage history. Mixed genotypes were assigned on the basis of IUPAC nucleotide ambiguity codes; for instance, the triplet GRT could refer to GAT or GGT, corresponding to amino acids aspartic acid (D) and glycine (G), respectively. We occasionally observed the triplet RRT, which

could correspond to a mix of aspartic acid (D; GAT), glycine (G; GGT), asparagine (N; AAT), and serine (S; AGT). We chose to annotate triplet RRT as a mix of D, G, and N, given that this mixed population has been previously observed by multiple groups (*Tamura et al., 2013*; *Mishin et al., 2014*; *Mohr et al., 2015*), whereas serine is two mutations away from the D consensus identity and is not present in the H3N2 GISAID sequences that we analyzed.

Passage histories are not recorded in a standardized fashion and are frequently missing altogether. In parsing the passage histories of isolates in the EpiFlu database, therefore, we sought only to sort sequences into broad categories of which we could be reasonably certain: egg-passaged, cell-culture-passaged, and unpassaged isolates. For instance, sequences with passage annotations containing 'MDCK,' 'SIAT,' 'RHMK,' 'MEK,' and various other cell-culture signifiers were combined into the broad category of cell-culture-passaged isolates. Our exact parsing procedures and the computer code used for analysis are available in *Figure 1—source data 1*.

## Viral strains

HA and NA sequences from the A/Brisbane/10/2007 (H3N2) strain were cloned into the bidirectional pHW2000 backbone to generate virus by reverse genetics (*Hoffmann et al., 2000*). We performed site-directed mutagenesis on the HA and NA to match the amino-acid (but not nucleotide) sequence from A/Hanoi/Q118/2007 (Genbank accessions AEX34134 and AEX34137 for the HA and NA, respectively), which has a G at NA site 151 (*Bao et al., 2008*). The HA and NA protein sequences are identical to those from A/California/UR06-0565/2007 (Genbank accessions ABW40191 and ABW40194 for HA and NA, respectively) aside from a single site in HA, as well as the genotype at NA site 151. We performed further rounds of site-directed mutagenesis to generate the D151 and N151 variants of the NA. The HA sequence from the A/Wisconsin/67/2005 (H3N2) influenza strain (Genbank accession CY163744) was similarly cloned into the bidirectional pHW2000 backbone for reverse genetics. The binding-deficient HA is derived from the A/Hong Kong/2/1968 (H3N2) HA and contains extensive mutations and deletions that eliminate receptor-binding activity; this is the variant referred to as the 'PassMut HA' in (*Hooper and Bloom, 2013*). Coding sequences for the HA and NA genes used in this study are available in *Supplementary file 1*.

The remaining six viral genes were expressed from bidirectional reverse-genetics plasmids derived from the A/WSN/33 strain (pHW181-PB2, pHW182-PB1, pHW183-PA, pHW185-NP, pHW187-M, and pHW188-NS) and were kind gifts from Robert Webster of St. Jude Children's Research Hospital. For all experiments not otherwise indicated, we used a plasmid (PB1flank-eGFP) that carried GFP flanked by PB1 packaging signals in place of pHW182-PB1 plasmid, and propagated the viruses in 293T and MDCK-SIAT1 cells expressing the WSN PB1 under the control of a CMV promoter as described in (*Bloom et al., 2010*).

## Viral reverse genetics

To generate GFP-expressing virus using reverse genetics, we transfected co-cultures of 293T-CMV-SIAT-PB1 and MDCK-SIAT1-CMV-PB1 cells with plasmids encoding the eight viral genes, with PB1flank-GFP rather than PB1 as described in (*Bloom et al., 2010*). We plated 2 x 10$^5$ 293T-CMV-PB1 cells and 0.2 x 10$^5$ MDCK-CMV-PB1 cells per well in six-well dishes in D10 (Dulbecco modified Eagle medium supplemented with 10% heat-inactivated fetal bovine serum [FBS], 2 mM L-glutamine, 100 U/mL penicillin, and 100 μg/mL streptomycin) and transfected each well with 2 μg plasmid DNA, corresponding to 250 ng of each of the eight plasmids, using the BioT transfection reagent (Bioland Scientific, Paramount, California). At 12 to 18 hr post-transfection, the cells were washed once with phosphate-buffered saline (PBS), and the media was changed to low-serum influenza growth media (IGM; Opti-MEM supplemented with 0.01% heat-inactivated FBS, 0.3% bovine serum albumin, 100 U/mL penicillin, 100 μg/mL streptomycin, and 100 μg/mL calcium chloride). TPCK (toylsulfonyl phenylalanyl chloromethyl ketone)-trypsin was added to IGM at 3 μg/mL immediately before use. For reverse genetics carried out in the presence of oseltamivir, we added the indicated concentration of oseltamivir carboxylate (kindly provided by Roche) to the IGM at this point as well. We collected viral supernatant at 72 hr post-transfection, clarified by centrifugation at 285xg for 4 min, aliquoted, and froze at -80 degrees C before thawing aliquots for titering. To generate viral populations that expressed the A/WSN/33 PB1 gene rather than the PB1 segment packaging GFP, we substituted 293T and MDCK-SIAT1 cells for 293T-CMV-PB1 and MDCK-SIAT1-CMV-PB1 cells in the protocol above.

## Viral titering

For viruses grown with the PB1flank-eGFP gene, titers were determined using flow cytometry. We plated $10^5$ MDCK-SIAT1-CMV-PB1 cells per well in 12-well plates in IGM and infected them 4–6 hr later with 0.1, 1, 10, or 100 μL of viral supernatant. At 16 hr post-infection, we collected the cells into PBS with 1% paraformaldehyde from wells in which approximately 1–10% of cells were GFP-positive and used flow cytometry to determine the exact proportion of GFP-positive cells. We used the Poisson equation to calculate the number of infectious particles in the original inoculum as:

[titer, in infectious particle per μL] = -log(1 – [fraction of GFP-positive cells]) * [number of cells plated, in this case $10^5$] / [inoculum volume, in μL]

For viruses grown with the WSN PB1 gene, titers were determined by staining for intracellular NP. Similar to the GFP titering described above, MDCK-SIAT1 cells were infected with serial dilutions of viral supernatant. At 12 hr post-infection, the cells were collected, fixed and permeabilized with the BD Cytofix/Cytoperm kit (product number 554722, BD Biosciences, Franklin Lakes, New Jersey) following the manufacturer's protocol but omitting the GolgiPlug, stained with a 1:20 dilution of mouse anti-NP FITC-conjugated antibody (clone A1 from MAB8257F, EMD MilliPore, Darmstadt, Germany), washed twice, and analyzed by flow cytometry to count NP-positive cells. The viral titer was computed from the fraction of positive cells using the Poisson equation above. All titers are plotted with the lower bound of the y-axis set at the limit of detection of this assay, approximately $10^{-1}$ infectious particles/μL.

Note that all of these titering methods quantify the number of virions that enter cells and express a functional polymerase complex that produces large amounts of the mRNA encoding the protein product being detected (GFP or NP). Unlike in a TCID50 or plaque assay, not all detected virions are necessarily able to undergo multi-cycle infections.

## Viral serial passage in cell culture

For each passage, we plated $10^5$ MDCK-SIAT1-CMV-PB1 cells per well in six-well plates in IGM and infected them 4–6 hr later with D151 viruses, G151 viruses, or an equal mix of the two at a total MOI of 0.2. An hour after the viruses were added, we washed the cells with PBS and added fresh IGM supplemented with 3 μg/mL TPCK-trypsin to dilute the effect of any oseltamivir remaining from reverse genetics for the first passage. We collected viral supernatant at 40 hr post-infection, clarified by centrifugation at 285xg for 4 min, aliquoted, and froze it at -80 degrees C before thawing aliquots for titering. We collected cells remaining at the end of each passage in 1mL Trizol reagent and froze them at -20 degrees C. From passage 4 onwards, mutation accumulation in the PB1flank-eGFP gene caused widespread loss of GFP in many viral populations; by the end of this passage, cytopathic effect was clearly visible even though GFP fluorescence was not. To inoculate passage 5, we infected cells with 5uL viral supernatant from passage 4, a volume corresponding approximately to an MOI of 0.2 based on the titers for earlier passages.

## Targeted deep sequencing of the NA gene

We extracted RNA from cells remaining at the end of each passage using Trizol reagent and performed reverse-transcription using the primers CAGGAGTGAAAATGAATCCAAATCAAAAGATAATAACGATTG and TTGCGAAAGCTTATATAGGCATGAGATTGATG, which target the full-length NA gene. We then used primers CTTTCCCTACACGACGCTCTTCCGATCTxxxCAACACTAAACAACGTGCATTCAAATGAC and GGAGTTCAGACGTGTGCTCTTCCGATCTCCTAACTCATTCATCAATAGGGTCCGATAAGG to amplify a targeted region of the NA gene surrounding site 151 and add the first half of the Illumina sequencing adaptor and a three-mer in-read barcode, represented here as xxx. We performed 25 cycles of amplification at an annealing temperature of 55 degrees C and an extension time of 40 s. We purified the PCR product using 1.5X Ampure beads and used this product as template for a second round of PCR using primers AATGATACGGCGACCACCGAGATCTACACTCTTTCCCTACACGACGCTCTTCC and CAAGCAGAAGACGGCATACGAGATxxxxxxGTGACTGGAGTTCAGACGTGTGCTCTTCC, which add the second half of the Illumina sequencing adaptors and a six-mer barcode, represented here as xxxxxx.

To sequence the viral stocks used to inoculate the first passage (these inoculating stocks are referred to as 'Passage 0' in *Figure 3*), we plated $10^5$ MDCK-SIAT1-CMV-PB1 cells per well in six-well plates in IGM and infected them 4–6 hr later with viral stocks at an MOI of 0.02. An hour after

the viruses were added, we washed the cells with PBS and added fresh IGM to dilute the effect of any oseltamivir remaining from the generation of virus by reverse genetics. We collected the cells in 1mL Trizol reagent at 16 hr post-infection and froze them at -20 degrees C. The purpose of this inoculation was to ensure that sequencing of viral stocks detected only infectious particles. We then prepared PCR amplicons from these samples for deep sequencing as described above.

Reads were first screened to verify sequencing quality and correct identity. Reads were discarded if any position had a Q-score below 25 or if the read had more than 4 mismatches relative to the plasmid reference sequence. We translated the reads in the NA reading frame and tallied the amino-acid identities at each position, recording "X" for positions with a discrepancy between the forward and reverse reads. The FASTQ files and the computer code used to analyze them are available in *Figure 3—source data 1*.

## Acknowledgements

We thank Roche for providing the oseltamivir carboxylate used in our experiments, Juhye Lee for cloning the A/Wisconsin/67/2005 hemagglutinin gene, Ian Gong for assistance with preliminary work, Choli Lee and Seungsoo Kim for assistance with sequencing, and Michael Emerman and Wenying Shou for helpful comments on the manuscript. This work was supported by the NIGMS of the NIH under grant R01GM102198. KSX was supported by an NSF Graduate Research Fellowship under grant number DGE-1256082 and a fellowship from the Fannie and John Hertz Foundation. KAH was supported by NRSA training grant T32GM007270.

## Additional information

### Funding

| Funder | Grant reference number | Author |
| --- | --- | --- |
| National Institute of General Medical Sciences | R01GM102198 | Jesse D Bloom |
| National Science Foundation | DGE-1256082 | Katherine S Xue |
| Hertz Foundation | Hertz Graduate Fellowship | Katherine S Xue |
| National Institute of General Medical Sciences | T32GM007270 | Kathryn A Hooper |

The funders had no role in study design, data collection and interpretation, or the decision to submit the work for publication.

### Author contributions

KSX, KAH, Conception and design, Acquisition of data, Analysis and interpretation of data, Drafting or revising the article; ARO, Acquisition of data, Analysis and interpretation of data, Drafting or revising the article; ASD, Conception and design, Analysis and interpretation of data, Drafting or revising the article, Contributed unpublished essential data or reagents; JDB, Conception and design, Analysis and interpretation of data, Drafting or revising the article

### Author ORCIDs

Katherine S Xue, http://orcid.org/0000-0002-4094-3615
Jesse D Bloom, http://orcid.org/0000-0003-1267-3408

## Additional files

### Supplementary files

- Supplementary file 1. This text file contains the coding sequences for the HA and NA genes used in this study.

## Major datasets

The following dataset was generated:

| Author(s) | Year | Dataset title | Dataset URL | Database, license, and accessibility information |
|---|---|---|---|---|
| Xue KS, Hooper KA, Ollodart AR, Dingens AS, Bloom J | 2016 | Data from: Cooperation between distinct viral variants promotes growth of H3N2 influenza in cell culture | http://dx.doi.org/10.5061/dryad.s3rs0 | Available at Dryad Digital Repository under a CC0 Public Domain Dedication |

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
