## [Decision Letter]

Thank you for submitting your work entitled "Cooperation between distinct viral variants promotes growth of H3N2 influenza in cell culture" for consideration by *eLife*. Your article has been favorably evaluated by Diethard Tautz (Senior Editor) and three reviewers, one of whom is a member of our Board of Reviewing Editors. Two of the three reviewers have agreed to reveal their identity: Marco Vignuzzi and James Paulson.

The reviewers have discussed the reviews with one another and the Reviewing Editor has drafted this decision to help you prepare a revised submission.

Summary:

The manuscript by Bloom et al. investigates cooperation between two variants of the influenza A virus neuraminidase proteins. Cultures of mixtures of D151 and G151 grow to higher titers than either variant alone, each variant invades a pure culture of the other and the mix settles at intermediate frequencies. Virus called as ambiguous at these sites have become common in Genbank since 2007. The 151 mutation rescues HA mutants incapable of binding to sialic acid and the enhancement is only observed for HA variants later than 2007 and not in 2005, suggesting that mutations in HA affecting the ability to the host receptor has made the mixed infections advantageous.

Essential revisions:

We would like to understand how cooperation depends on the multiplicity of infection. MOIs of 0.2 and 0.5 seem to behave similarly and experiments at lower MOI would be instructive.

A large fraction of the influenza sequences in data bases are annotated with passage information. It would be informative to stratify the sequences by passage and redo Figure 1 for different passage categories. Direct sequencing data of RNA from patient isolates could shed light on the existence of mixed populations in clinical samples. If such data were available, this could further strengthen this paper.

*Reviewer #2:*

While the authors included the Bordería et al. reference in the Introduction, I believe that to be fair, they should refer more directly to this work in the Discussion where they state that their own work demonstrates that cooperation between distinct variants can enhance overall fitness. In Bordería et al., the authors showed that variants of several residues on different structural proteins also emerged as a mixed population whose total fitness was greater than any single virus population alone – and these mutants also affect receptor binding.

---

## [Author Response]

Summary:

*The manuscript by Bloom et al. investigates cooperation between two variants of the influenza A virus neuraminidase proteins. Cultures of mixtures of D151 and G151 grow to higher titers than either variant alone, each variant invades a pure culture of the other and the mix settles at intermediate frequencies. Virus called as ambiguous at these sites have become common in Genbank since 2007. The 151 mutation rescues HA mutants incapable of binding to sialic acid and the enhancement is only observed for HA variants later than 2007 and not in 2005, suggesting that mutations in HA affecting the ability to the host receptor has made the mixed infections advantageous.*

We thank the editors and reviewers for their careful review of our study. We have carried out several additional experiments and analyses that we believe have greatly improved the manuscript. These revisions are summarized here and described at greater length below. Briefly:

a) We have tested the effect of cooperation at a low multiplicity of infection by inoculating infections at an MOI of 0.02 and assaying viral titer;

b) We have parsed the passage history metadata for sequences in the GISAID EpiFlu database to better understand how cooperation relates to passage history;

c) We have expanded the discussion of our results to better set them in the context of prior work suggested by the reviewers.

Essential revisions:

*We would like to understand how cooperation depends on the multiplicity of infection. MOIs of 0.2 and 0.5 seem to behave similarly and experiments at lower MOI would be instructive.*

This was a very helpful suggestion. We measured the growth of mixed and pure viral populations at an MOI of 0.02, and we serially passaged replicate pure and mixed populations at the same MOI. As the reviewers may have suspected, we find that at low MOI there is initially a lag period where the mixed population grows slightly worse than the pure D151 population. But by later times, we find that mixed populations grow better than pure populations even at an MOI of 0.02. We proposed that cooperation occurs when the D151 and G151 viruses co-infect the same cell, which would occur less frequently during the initial stages of a lower MOI infection. In accordance with these predictions, we see that the mixed populations grow slightly worse than the pure populations during the first 24 hours of the passage, when the MOI is low. Later in the passage, when the MOI increases due to accumulation of virus in the supernatant, we find that the mixed populations grow better than the pure populations. These new data are what in now is Figure 4 of the revised paper.

*A large fraction of the influenza sequences in data bases are annotated with passage information. It would be informative to stratify the sequences by passage and redo Figure 1 for different passage categories. Direct sequencing data of RNA from patient isolates could shed light on the existence of mixed populations in clinical samples. If such data were available, this could further strengthen this paper.*

This was another helpful suggestion that helped us set cooperation in the context of passage protocols and clinical infections. We parsed the passage annotations from sequences in the GISAID EpiFlu database and observed mixed populations almost exclusively in strains that had been passaged in cell culture. Notably, mixed populations were detected in only two of the unpassaged isolates. These new data are included in the revised Figure 1.

However, as we note in the Discussion, these sequences are obtained through Sanger sequencing, which may not detect G151 if it is a rare variant in the population. For instance, a number of Sanger sequencing studies report G151 as “fixed” in cell culture passaged virus – something that appears very unlikely in light of our experimental results. It therefore remains unclear of G151 arises fully de novo in cell culture, or is simply enriched from pre-existing variation.

Certainly, as the reviewers suggest, deep sequencing of unpassaged clinical isolates is the only way to definitively determine whether cooperation occurs in clinical infections. However, this work requires access to clinical samples that place it outside the scope of our current study. In any case, we note that the identification of viral cooperation even in cell culture is a finding of significant evolutionary interest – particularly since it arises so repeatedly and consistently. As is often the case in biology, we hope that our characterization of this fascinating phenomenon in the lab will provide a scientific motivation for further studies that examine if the same principles apply in other settings.

*Reviewer #2: While the authors included the Bordería et al. reference in the Introduction, I believe that to be fair, they should refer more directly to this work in the Discussion where they state that their own work demonstrates that cooperation between distinct variants can enhance overall fitness. In Bordería et al., the authors showed that variants of several residues on different structural proteins also emerged as a mixed population whose total fitness was greater than any single virus population alone* – *and these mutants also affect receptor binding.*

This was another helpful suggestion. The work of Bordería et al. bears several similarities to ours, and we have noted the connections and context in both the Introduction and Discussion. We hope these revisions provide better context for readers.